# The Effect of Burnout Experienced by Nurses in Retirement Homes on Human Resources Economics

Ljiljana Leskovic [1], Sergej Gričar [2], Raffaella Folgieri [3], Violeta Šugar [4,*] and Štefan Bojnec [5,6]

1   Department of a Holistic Patient Care, Faculty of Health Sciences, University of Novo Mesto, Na Loko 2, 8000 Novo Mesto, Slovenia; ljiljana.leskovic@uni-nm.si
2   Department of Quantitative Economics, Faculty of Business and Management Sciences, University of Novo Mesto, Na Loko 2, 8000 Novo Mesto, Slovenia; sergej.gricar@uni-nm.si
3   Department of Philosophy, University of Milan, Via Festa del Perdono 7, 20122 Milano, Italy; raffaella.folgieri@unimi.it
4   Department of Entrepreneurship and Quality, Faculty of Economics and Tourism "Dr. Mijo Mirković", Juraj Dobrila University of Pula, Zagrebačka 30, 52100 Pula, Croatia
5   Department of Economics, Faculty of Management, University of Primorska, Izolska Vrata 2, 6000 Koper, Slovenia; stefan.bojnec@fm-kp.si
6   Department of Economics, Faculty of Economics and Management, Czech University of Life Sciences Prague, Kamýcká 129, 165 00 Prague, Czech Republic
*   Correspondence: violeta.sugar@unipu.hr

**Abstract:** The human resources economic implications of nursing burnout amongst nurses working in retirement homes have become a critical concern within the healthcare industry. As the backbone of care provision in these settings, it is crucial to understand the consequences of burnout on the workforce's well-being and organisational sustainability. This study aims to investigate burnout among nurses working in retirement homes in Slovenia. The reasons for burnout vary across countries and regions, so gathering data specific to this population is essential. Through surveys conducted among 253 nurses and medical technicians, factor analysis revealed three factors for burnout: emotional exhaustion, reduced personal fulfilment, and impersonality. This research aims to pave the way for reducing workplace stress by creating new opportunities for better working conditions. To achieve these goals, executive management in retirement homes should gain proficiency in the four elements of the quality management cycle: planning, execution, evaluation, and continuous improvement. Furthermore, a comparative analysis was conducted to collate the empirical findings with those from Croatia.

**Keywords:** quality of work; burnout; Croatia; factor analysis; management cycle; public sector economics; Slovenia

## 1. Introduction

Burnout is a serious issue resulting from a poor quality of work life (QWL), causing employees to feel exhausted, emotionally detached, and unproductive (Ashrafi et al. 2018). These issues can lead to decreased job performance and have significant costs for organisations. Burnout is a severe psychological syndrome that occurs due to prolonged emotional or psychological stress. It is characterised by three dimensions: emotional exhaustion, depersonalisation, and a reduced sense of personal accomplishment at work. Burnout affects 13–25% of the working population, with increasing costs. Therefore, there is a need for further research to understand the role of (de)motivators and their impact on work performance. QWL is a crucial constraint component of organisational performance. This study aims to broaden the understanding of the impact of professional burnout on nurses working in retirement homes. International studies (Woo et al. 2020) have shown that burnout rates among nurses range from 30% to 80%. It is essential to address this issue to ensure the well-being of nurses and the quality of care provided to elderly residents.

High staff turnover rates in healthcare systems severely challenge policymakers, healthcare providers, and healthcare users or patients. Regularly providing healthcare services increases the cost of hiring, positioning, and training and the workload of employees on duty. Overall, burnout affects the quality of work in retirement homes significantly. Nurses play a central role in providing care to residents, and burnout can compromise the empathetic connection between caregivers and residents, leading to suboptimal patient outcomes. The negative cycle perpetuates due to increased absenteeism and turnover rates, disrupting the continuity of care. Recognising and mitigating the impact of burnout is essential for fostering a healthcare environment that prioritises human resources and those entrusted to their care.

In management planning, the acceptable burden of employees is one of the most notable, meaningful, and issuable elements that play a significant role in private or public businesses, companies, or health organisations (e.g., retirement homes). Therefore, the knowledge and capability of managers to correctly designate the amount of work to their employees is paramount, and managers must adopt strategies that result in a financially, socially, and employee-friendly environment (Černe et al. 2014). This is important for non-tradable services for which national incomes and labour capitalisation generate benefits. The nursing process in retirement homes is under a pressing need for employees; users expect extensive and intensive services produced individually and without robots or similar automated services that do not include the necessary social interactions. Good services can advance any organisation's development; however, providing good services causes much pressure upon employees when the working process is not optimised.

Regardless of whether the service sector is public, private, or non-governmental (e.g., an NGO), the environment and design of contemporary retirement homes deserve more attention and scientific research in different fields, such as economics (demand), management (knowledge of employees and leaders or chief executive officers (i.e., CEOs)), social interactions (with service stakeholders), and psychosocial working conditions (the definitions of burnout).

Considering this and the third element of the management Deming cycle (i.e., "check") (Koiesar 1994), the goal of this paper was to study and determine the incidence of burnout among nurses in Slovenian retirement homes and to examine the interactions between influential factors that cause burnout. Surveys were conducted in 2017 among approximately 8.7% of all institutions at nursing care providers in retirement homes in Slovenia. This paper aims to provide significant knowledge and in-depth insight into burnout among nurses, which could enable managers to respond appropriately for both employees (nurses) and patients, that is, service users. This is also relevant for better service quality for various stakeholders. This research underscores the vital role of informed management in creating a supportive and sustainable work environment. The recent theoretical academic literature shows that healthcare professionals are particularly susceptible to burnout due to risky decision-making involving people, empathy, and responsibility (Lluch et al. 2022). The literature addressing burnout among nurses in retirement homes, where specialised services are tailored to individuals typically aged 80 and above (Friganović et al. 2019; Lund et al. 2023), is noticeably sparse, especially when juxtaposed with the extensive research conducted on various aspects of nursing practice (De Diego-Cordero et al. 2022) and investigations into COVID-19 prevalence rates (Mary Pappiya et al. 2023), but not on demographic issues (Sherman and Touhy 2017). Consequently, there is an imperative to establish conducive conditions for nursing staff to flourish and grow, ensuring their capacity to provide high-quality care. This contrasts with working environments where resource deficiencies contribute to burnout, prompting individuals to exit the profession or adapt to suboptimal care standards (White et al. 2019).

A shortage of qualified nurses is a critical challenge in healthcare worldwide (Bratt and Gautun 2018). Staff shortage is a multidimensional phenomenon attributed to low job satisfaction, lack of managerial support, and poor career opportunities (Atefi et al. 2015; Hayes et al. 2010; Meeusen et al. 2024). Several authors (Dimec et al. 2008; Dimond 2004;

Glasberg et al. 2007; Selič et al. 2010) point out that burnout is a sign of exhausted and defective bio-psycho-social well-being, leading to absenteeism or leaving the profession. Amongst the fundamental causes of stress in recent years, the extension of working hours prevails and causes great suffering to the entire family (Premeaux et al. 2007).

The balance between work and quality of life for nursing professionals in retirement homes is delicate. Unfortunately, burnout syndrome disrupts this equilibrium, causing emotional exhaustion, depersonalisation, and reduced personal fulfilment. This not only affects nurses' satisfaction with their job but also their personal life. Burnout can cast a shadow on personal relationships and overall life satisfaction. It is crucial to address burnout to maintain an efficient organisation and preserve the overall well-being of healthcare workers.

Burnout also has considerable economic implications for healthcare organisations (West et al. 2018). Increased absenteeism and turnover rates necessitate additional investments in recruiting and training new staff, adding financial strain to the human resources economics within the sector (Gruzina et al. 2021). Furthermore, workplace inefficiencies resulting from burnout can disrupt organisational productivity, translating into economic challenges that healthcare facilities must navigate (Leitão et al. 2021). Therefore, it is crucial to implement strategies that enhance the overall well-being of nursing professionals, foster supportive work environments, and strategically mitigate the economic implications of burnout. This will contribute to sustainable and effective healthcare delivery in retirement homes, restoring the symbiotic relationship between the quality of life and work (Aiken 2008).

This study aims to contribute significantly to investigating the underexplored realms of nursing science and management within retirement homes. There are four primary objectives of this study. Firstly, it seeks to consolidate knowledge on burnout syndrome by reviewing findings from previous empirical research, providing a comprehensive overview of the current state of understanding in the field. Furthermore, it aims is to conduct a comparative analysis of the research outcomes with those obtained from the adjacent country of Croatia, thus offering valuable insights for prospective research avenues. Secondly, this study emphasises the scientific contribution to the performance of burnout among nursing employees, specifically within the context of retirement homes, recognising the unique challenges and dynamics of this setting. Thirdly, this research delves into the analysis of individual data obtained through surveys, employing factor analysis to uncover the underlying dimensions of burnout among nursing staff in retirement homes. This applied analytical approach adds a nuanced layer to the existing literature by elucidating specific factors contributing to burnout in this distinct healthcare environment. Finally, this study goes beyond data analysis to present results and findings, offering insights into the significant aspects of burnout syndrome among employees in retirement homes. The discussion of these findings extends to their practical implications for policy and practice, providing evidence-based, actionable insights for improving working conditions and mitigating burnout in these critical healthcare settings.

## 2. Literature Review

Nursing is a profession characterised by interactions with individuals requiring a high degree of service. Women are also burdened because they are torn between their careers and family responsibilities (Lan et al. 2020). The scientific literature on nursing shows a lack of research regarding burnout syndrome among nursing employees in Slovenia (Starc 2018; Rožman et al. 2017) and worldwide (Mengist et al. 2021; Savage et al. 2022; Sheets 2023).

The most recent study utilising factor analysis concerning nursing by Gurková et al. (2020) investigated a sample of almost 1500 participants. Additionally, other research developed the methodology mentioned earlier in nursing studies on psychometrics (Gusar et al. 2021), knowledge (Chen et al. 2022), and healthcare teaching (Tehranineshat et al. 2022; Al-Rawajfah et al. 2022), but rarely in retirement homes, which constitutes our state-of-the-art research study (Zhou et al. 2022).

In management, planning weekly or daily nursing activities is of utmost importance (Van Bogaert et al. 2013). Moreover, employees' work should be studied regarding the third management cycle element (Tamher et al. 2021; Baier et al. 2004) (C or S: check/study); therefore, Zeleníková et al. (2020) studied this phenomenon of unfinished nursing work, which means that nurses did not provide appropriate care for a patient/person. The study was performed in four central European countries. Their findings and conclusions could be essential for explaining and understanding the correlation between planning issues and burnout syndrome. A previous study discussed management responsibilities for nursing problems and solving these issues (Ujoatuonu et al. 2023). Other relevant contemporary studies on nursing burnout in different sectors of medical treatments include clinical nurses (Chachula 2021; Xie et al. 2021; Patrician et al. 2022), foreign countries, e.g., Vietnam (Tran et al. 2023), and a meta-analysis on satisfaction and burnout prevalence (Algamdi 2022).

Research on burnout among nurses in retirement homes was performed in 2002 (Evers et al. 2002), and the study was based on 551 questionnaires. Their findings are relevant, as they recognised burnout as a three-dimensional syndrome of each person involved in nursing activities in retirement homes, regardless of sex or age. This syndrome could be quantified and was significantly related to working hours per week. Generating a sufficiently large workforce and stable environment in nursing healthcare facilities can substantially influence and reduce burnout among nurses. This aligns with a subsequent study (Narumoto et al. 2008) in Japan and could be implemented in the Slovenian working environment. Furthermore, the authors (Irinyi et al. 2019) reported not only that employees are susceptible to burnout but also that management at all three manager levels (top, middle, and low levels) can be affected by management burnout (Laschinger et al. 2004). Middle managers reported more incredible burnout than employees (Hewko et al. 2015).

Little attention is paid to burnout among nurses in retirement homes in the literature (French et al. 2022; Benbow 1998; Demerouti et al. 2000). Consequently, this gap in the literature is addressed by our current study, which involves conducting in-depth surveys and analyses to yield valuable and pertinent results and findings. Our investigation aims to ascertain whether burnout among nurses is significantly associated with specific syndromes.

To sum up, the empirical literature reveals a significant gap in the research on burnout syndrome among nursing employees globally and in Slovenia. This study emphasises the importance of addressing this gap and discusses the research on nursing burnout and its impact on employees and management levels. To fill the literature gap, this study aims to conduct thorough surveys on burnout among nurses in retirement homes. The main objective is to explore the associations with specific syndromes and compare the results with previous studies. This study highlights the crucial balance between work and quality of life for nursing professionals in retirement homes and stresses the negative impact of burnout on emotional well-being. It also discusses the economic implications of burnout, such as increased absenteeism and turnover rates, and emphasises the need for strategies to enhance overall well-being, create supportive work environments, and mitigate economic challenges in healthcare organisations.

## 3. Materials and Methods

### 3.1. Study Design

The survey in this study was conducted in ten retirement homes in Slovenia, accounting for 8.7% of all retirement homes in the country. The data were collected through a questionnaire that was custom-designed and validated. The questionnaire included the Maslach burnout inventory human services survey (MBI-HSS), the job descriptive index (JDI), questions related to demographic factors, and the psychosomatic disorders checklist. The MBI-HSS scores were analysed in three dimensions: emotional exhaustion, depersonalisation, and personal accomplishment. For the study, one retirement home was selected from every Slovenian region, and 50 questionnaires were sent to the management of each

retirement home. We asked them to survey with the enclosed questionnaires among the employees involved in medical treatment (nurses and medical technicians).

### 3.2. Outcome Measures

Factor analysis was used for the data analysis. Principal axis factoring (direct oblimin rotation) was used to check the structure of individual questionnaire sets and the reliability of the instruments. We verified the internal consistency by calculating Cronbach's coefficient $\alpha$. Categorical items were described with absolute and relative frequencies, and numerical items were described with arithmetic means, standard deviations, medians, minimum and maximum values, and normalities (skewness and kurtosis). The normality of distribution was verified by the Kolmogorov–Smirnov and Shapiro–Wilk tests. A nonparametric Spearman correlation coefficient confirmed the correlation between variable pairs, and the differences between the two independent groups were verified by the Mann–Whitney test. Relationships and differences were tested at a 5% significance level. The collected data were processed in the IBM Statistical Package for Social Science, (SPSS) version 21.0 (IBM, North Harbour Portsmouth Hampshire, UK).

The International Council of Nurses noted that creating a positive working and living environment and supporting good practices in the health sector are essential to maintaining patient safety and the well-being of healthcare workers. The burnout process takes place in successive stages, from the state of exhaustion to the state of captivity and further to the state of adrenal burnout, and this process very much depends on the personality traits of every individual (Parola et al. 2022).

Changing working conditions, transitions of job responsibilities, an ageing population, and globalisation in the European Union (EU) job market are causing increased psychological stress (Berényi 2022). Nursing personnel in retirement homes are exposed to several factors that may contribute to burnout syndrome (Rizo-Baeza et al. 2018). Identifying the risk factors for burnout syndrome is crucial for countries, individuals, and organisations to develop effective strategies to reduce and prevent burnout among their employees (Moss et al. 2016). Therefore, the motivation of our research is to comprehensively investigate and ascertain the prevalence of burnout syndrome among the nursing staff employed in retirement homes situated in Slovenia. Beyond simply identifying the prevalence, we are committed to delving deeper into the intricate web of factors that may be associated with this syndrome within the context of Slovenian retirement homes. By doing so, we seek to shed light on the nuanced dynamics and potential determinants of burnout in this healthcare setting. Our study is driven by the recognition that understanding these factors is essential for the nursing staff's well-being and for enhancing the quality of care provided to the elderly residents in these retirement homes. Through rigorous surveys, analyses, and data interpretation, we aim to contribute valuable insights that can inform policies, interventions, and strategies to mitigate burnout and create a more supportive and sustainable work environment for nursing staff, ultimately benefitting healthcare professionals and the elderly population they serve.

### 3.3. Data Analysis

To measure burnout syndrome, the authors (Maslach et al. 2001) used the MBI-HSS to assess three aspects of burnout: emotional exhaustion, depersonalisation, and personal fulfilment. This scale consists of 22 items, measured on both dimensions (frequency and intensity) using two Likert-type scales. The MBI-HSS exhibits high consistency and reliability at approximately 90%. The data used in the questionnaire can reveal relevant variables corresponding to burnout aspects. All 22 items are presented. The survey was conducted with the participation of 253 individuals, among whom 41.5% were between the ages of 40 and 49. The sample size used a standard 95% confidence level and a 5% risk of

the specified interval, not including the population value. The calculation was performed using the following equation:

$$sample\ size = \frac{z^2 \cdot p(1-p)}{e^2} / 1 + \frac{(z^2 \cdot p(1-p))}{e^2 \cdot N} = 180, \tag{1}$$

where *n* is the sample size, *z* = constant for a 95% confidence level, *p* is the expected error rate, and *ε* is the precision level. The required sample size, in our case, was at least 180 units. Our sample included 253 respondents, so it exceeded the minimum requirement.

## 4. Results

The results are presented in Table 1 according to emotional exhaustion, depersonalisation, and reduced personal accomplishment. The frequency of occurrence was estimated using values from 1 to 6, from rare observation (a few times per year) to frequent observation (every day). The intensity was denoted with values 1 to 7, from an abysmal experience to a solid background. In both cases, the value 0 could be chosen if the state had never been experienced. Due to stringent restrictions on accessing extensive data, the collection of sociodemographic information from the participants was limited in scope.

**Table 1.** Pattern matrix MBI-HSS questionnaire.

| No. | Variable (Questionnaire Item) | Factor | | |
|---|---|---|---|---|
| | | 1 | 2 | 3 |
| 1 | At the end of work, I feel worn out. | 0.87 | | |
| 2 | I feel burnt out from work. | 0.83 | | |
| 3 | I feel tired when I get up to face a new workday. | 0.80 | | |
| 4 | I feel like I have no energy. | 0.78 | | |
| 5 | My work emotionally drains me. | 0.76 | | |
| 6 | My work frustrates me. | 0.70 | | |
| 7 | Working with people all day is tiring for me. | 0.69 | | |
| 8 | I get the feeling that I am doing too much. | 0.60 | | |
| 9 | Working with people results in too much stress for me. | 0.49 | | |
| 10 | I am effectively coping with my patient's problems. | | 0.77 | |
| 11 | I quickly established a relaxed atmosphere in patient relationships. | | 0.66 | |
| 12 | I feel full of energy. | | 0.57 | |
| 13 | It is easy for my patients to understand what and how they feel. | | 0.53 | |
| 14 | At work, I am very calm in dealing with emotional issues. | | 0.50 | |
| 15 | I have done many useful things at work. | | 0.49 | |
| 16 | I feel that I have a positive impact on people through my work. | | 0.48 | |
| 17 | I am happy when I work with patients. | | 0.42 | |
| 18 | I treat certain patients as impersonal objects. | | | 0.91 |
| 19 | I feel that patients blame me for their problems. | | | 0.62 |
| 20 | I have become colder to people since starting this job. | | | 0.53 |
| 21 | I do not really care what happens to my patients. | | | 0.50 |
| 22 | I am worried that my job is making me emotionally difficult. | | | 0.47 |

The survey participants consisted of nurses and medical technicians from Slovenian retirement homes, of which 90.51% were female, 5.14% were male, and 4.35% did not state their gender. Regarding age, 8.7% of the respondents were <30, 18.2% were between 30–39, 41.5% were between 40–49, 29.2% were in the age group 50–59, and 2.4% were >60. Regarding shift work, 24.90% worked one shift, 18.98% worked two changes, and 56.12% worked three shifts.

The structural decomposition of the 22 variables was verified by factor analysis using the principal axis factoring method, and the direct oblimin method was used for rotation. The number of factors was determined by a scree plot, which clearly showed a connection by the third factor. The Kaiser–Meyer–Olkin measure of sampling adequacy was 0.824,

and Bertlett's test was statistically significant ($p < 0.001$), which means that the data in the sample were fit for factor analysis. Three isolated factors explained 45% of the total variability of the model (Table 1), where all the results are visible in Table A1.

The results revealed that the component loadings for each variable were >0.4. Since the variables on one factor had a weight factor of >0.4 and the difference between the factor weights on an individual item was >0.2, the convergence and discriminant validity of the measuring instrument was confirmed. Table 1 and Figure 1 show the factor solution with three factors. The first factor represents nine variables, i.e., the emotional exhaustion dimension of burnout among nurses. The second factor binds eight variables, referred to as the reduced personal fulfilment dimension of burnout among nurses. The third factor, with five variables, represents the impersonality dimension of burnout among nurses.

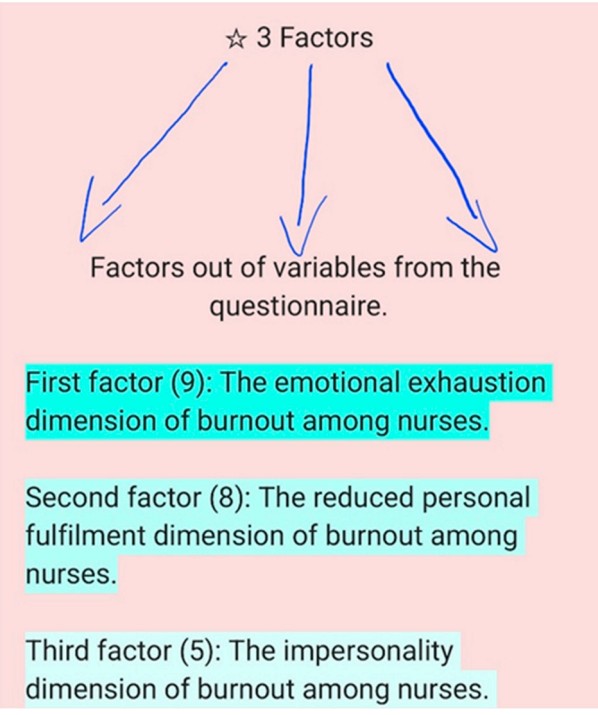

**Figure 1.** The factors of the research.

The reliability of the factor analysis results was verified by Cronbach's alpha coefficient, which was $\alpha = 0.911$ for factor 1, $\alpha = 0.754$ for factor 2, and $\alpha = 0.729$ for factor 3. For all three factors, the Cronbach alpha value was above 0.7, which confirms the method's reliability.

Three factors, now the average of the items that most represented a single dimension, were further analysed to check the normality of the results. The factors were calculated for the respondents who answered all the things that primarily represented a single factor. The normality of the distribution of the factors was checked with the Kolmogorov–Smirnov and Shapiro–Wilk tests, which revealed a statistically significant distribution for all variables ($p < 0.01$). Table 2 presents descriptive statistics of the composite variables representing different dimensions of burnout. The most common variable was emotional exhaustion (mean: 2.11), followed by reduced personal fulfilment (mean: 1.52), and the least frequent was impersonality (mean: 0.67).

Moreover, the results showed that all three components of burnout slightly differed between the genders. The level of statistical characteristics was between 0.174–0.716. Men, on average, exhibited greater emotional exhaustion and reduced personal fulfilment, whereas women showed more impersonality. Evaluating the three components of burnout based on age groups (<33 and >33 years of age) revealed significance levels between 0.226–0.558. Slightly higher fatigue in all three components was observed in individuals >33 years of age.

**Table 2.** Descriptive statistics of MBI-HSS composite variables.

| Item | Factor 1: Emotional Exhaustion | Factor 2: Reduced Personal Fulfilment | Factor 3: Impersonality |
|---|---|---|---|
| Mean | 2.11 | 1.52 | 0.67 |
| Median | 1.89 | 1.38 | 0.40 |
| Variance | 1.69 | 0.31 | 0.53 |
| Minimum | 0.11 | 1.00 | 0.00 |
| Maximum | 4.89 | 3.75 | 4.00 |

## 5. Discussion

Selič-Zupančič et al. (2023) demonstrated that women are more likely to experience burnout in all three psychological dimensions than men. A higher rate of burnout in women is attributed to the dual role women play in society; women are burdened with the upbringing of their children and running their households in addition to their work.

Several studies (Wynn 2013; Becker et al. 2006; Barrington et al. 2014; Wang et al. 2020) have demonstrated that nurses experience burnout during their careers, which has implications for their physical and mental health status. The costs due to nurses leaving their jobs or the entire profession could also generate financial crises and reduce the quality and availability of services for patients. Absenteeism is already foreseen and discussed by Chan et al. (2013). The safety and well-being of healthcare workers are not only a legal and social obligation but also a condition that brings many benefits to social institutions.

Bakker et al. (2002) assert that burnout syndrome is a potential risk for various healthcare professionals, among nurses, who also include technicians, physiotherapists, working therapists, social workers, and cleaners working in retirement homes. Moreover, these employees are essential for maintaining social welfare because they face the complexity of the health and social problems of prolonging the lives of the elderly population daily. This part of the population is also among those with the highest at-risk-of-poverty rate in Slovenia (Kabát et al. 2013; Gricar and Tomec 2023) and neighbouring Croatia (Vučemilović 2022), which creates additional economic, social, and emotional problems among older people and their family members. The development of gerontological care in Slovenia strongly influences the development of gerontological care in the western Balkans (Mijakoski et al. 2015; Amiri 2021), as they largely follow the practices in Slovenia due to their historical and economic proximity (Hirt and Ortlieb 2012; Kotsani et al. 2020). Therefore, it is crucial to understand burnout syndrome in healthcare institutions in Slovenia (Vladič and Kren 2021; Selič-Zupančič et al. 2023), as exemplary case studies and practices can be spread to neighbouring, particularly western Balkan, countries (Tucak Junaković and Macuka 2021).

Therefore, the first (sub)objective was to compare the findings from Slovenia with those previously conducted by other researchers in neighbouring Croatia (Neuberg et al. 2017; Bošković 2021), which found similar patterns. The comparative analysis between the results in Slovenia and those from Croatia revealed notable similarities and significant diversities in the experiences of burnout among nurses in retirement homes. Similarities were evident in both countries' shared prevalence of emotional exhaustion, depersonalisation, and reduced personal fulfilment dimensions of burnout. However, there were notable diversities in the contributing factors and their relative impacts. For instance, while workload was a prominent factor in both countries, its significance varied, with Slovenia emphasising the role of demographic characteristics, such as age and gender, and Croatia highlighting organisational factors (Cilar et al. 2021). These divergent patterns underscore the importance of considering the unique contextual factors influencing burnout experiences in different regions, ultimately guiding targeted interventions and strategies to alleviate burnout among nursing staff. More research is needed to thoroughly examine these subtleties and develop evidence-based approaches that fit specific healthcare environments (Rubic et al. 2022).

There is no unambiguous definition of burnout among nurses in retirement homes; however, most experts agree it is a chronic condition of extreme physical and emotional exhaustion. The results and findings of this study are thus novel and demonstrate that healthcare workers exhibit significant burnout syndrome, mainly regarding emotional exhaustion. Our research findings are consistent with Algamdi's (2022) professional quality of life (ProQOL) study. By comparing these two studies, we gain valuable insights into the complex landscape of burnout among healthcare professionals. Our research focused on nursing staff in retirement homes, while the ProQOL scale study focused on oncology nurses. Both studies highlight the multifaceted nature of burnout, with the Slovenian study exploring work-related dimensions and advocating for policy recommendations and continuous improvement.

Meanwhile, the ProQOL scale study provides a meta-analysis of compassion satisfaction (CS), burnout (BO), and secondary traumatic stress (STS). Both studies argue the significance of work-related factors and the need for tailored interventions. The Slovenian study emphasises economic and managerial implications, urging resource allocation and cost-effective interventions. On the other hand, the ProQOL scale study recommends specific interventions such as resilience-based programs and mindfulness exercises. While acknowledging limitations and advocating for future research, these studies contribute to a nuanced understanding of burnout, emphasising the importance of context-specific strategies to enhance healthcare professionals' well-being.

## 6. Conclusions

The main contribution of this study is to determine the prevalence of burnout syndrome among nursing staff working in Slovenian retirement homes and identify potential associated factors. This study used the MBI-HSS to measure burnout, which evaluated emotional exhaustion, personal fulfilment, and depersonalisation through a survey of 22 items. The survey included questions about the frequency and intensity of burnout-related feelings and sociodemographic information about the participants.

The participant pool primarily consisted of nurses and medical technicians, with the majority being females, while a smaller proportion were males, and some chose not to disclose their gender. Regarding age distribution, the participants were categorised into different age groups. Concerning shift work, the participants were divided into those working one, two, or three shifts.

A three-factor structure that explained a substantial portion of the variability in the sample was uncovered through factor analysis. The first factor was related to emotional exhaustion, the second to reduced personal fulfilment, and the third to impersonality. The reliability of these factors was validated using Cronbach's $\alpha$ coefficient, which consistently exceeded the acceptable threshold.

In summary, our study provided a comprehensive understanding of the prevalence of burnout syndrome and its dimensions among nursing staff in Slovenian retirement homes. The data collected contributed to our comprehension of the factors influencing burnout in this context and highlighted potential avenues for intervention and support. This study may initiate the process of improving the current hostile and exhausting situation for nurses with burnout syndrome in Slovenian retirement homes. Furthermore, our results contribute to the understanding of burnout syndrome. They can be used by ministries, national authorities, and social welfare institutions to evaluate the currently inadequate normative regulation of inhumane work and the lack of standards within different institutions. In addition, this study proposes plans to prevent burnout syndrome at the individual, organisational process, and employee education levels.

In light of the insights gained from our study, several policy recommendations emerge as crucial steps to address burnout among nursing staff in Slovenian retirement homes. First and foremost, there is a pressing need to establish and maintain supportive work environments within these facilities. This entails cultivating a culture of respect, offering resources for stress management, and ensuring adequate staffing levels to prevent excessive workloads, which are known contributors to burnout. Secondly, regular burnout screening and assessment tools should be integrated into retirement home practices. Moreover, policymakers and administrators should prioritise mental health support programs designed specifically for nursing professionals. Training and education in stress management, resilience building, and self-care strategies should be provided to empower nursing staff further. Supervisors and managers play a pivotal role in mitigating burnout. Therefore, their training should encompass the recognition of burnout signs and the ability to offer appropriate support and resources. Open communication channels between staff and management are essential for fostering a supportive work environment. Job roles within retirement homes should be reviewed and, if necessary, redesigned to reduce administrative burdens and allow more time for direct patient care, which can be fulfilling and satisfying. Staff development opportunities should be actively promoted to keep employees engaged and motivated.

Additionally, establishing a monitoring and evaluation system for burnout levels and intervention effectiveness is essential. Lastly, encouragement and support for research initiatives focused on burnout in retirement home settings are vital. This study's findings underscore the importance of incorporating elements from both the Deming Plan – Do- Check- Act (PDCA) cycle and the study/check phase of the PDCA cycle into managerial practices within retirement homes for human resources economics. By continuously planning, executing, monitoring, and analysing the impact of interventions aimed at reducing burnout, managers can foster a culture of proactive improvement, ensuring better working conditions and ultimately enhancing the well-being of nursing staff while delivering higher-quality care to residents.

This study has certain limitations that should be considered. For instance, the data used in this study may be subject to response bias, and the survey's sample size is relatively small. Given the relatively modest population of Slovenia and the limited number of healthcare practitioners available to serve retirement homes, a substantial sample size was deemed unnecessary.

Despite facing some limitations, it can be asserted with confidence that the sample employed in the study was dependable and practical. Nevertheless, it is imperative to mention that the specific demographic data and health status information of nurses and technicians engaged in retirement homes could not be collected due to ethical considerations. Furthermore, the diagnosis of certain medical conditions was strictly prohibited per Slovenia's regulations. Additionally, it is essential to remember that the study focused solely on Slovenian retirement homes and the specific burnout dimensions examined; therefore, it may not represent the broader spectrum of nursing staff experiences and contexts.

Future research endeavours could explore longitudinal data to establish causality in the relationships between burnout and associated factors among nursing staff in retirement homes. Comparative studies across different healthcare settings could impede more significant heterogeneity, and the limits gain a generalisability of the results. Moreover, the international contexts could provide valuable insights into variations in burnout prevalence and the contributing factors. The direct idea for future research is to conduct an identical survey in a specific country, e.g., Croatia, to have a direct comparative analysis of two neighbouring countries with similar health systems (Svitlić-Budisavljević 2022).

In conclusion, this study highlights the significant impact of demographic- and work-related factors on burnout among nursing staff in Slovenian retirement homes, underscoring the need for targeted interventions to mitigate burnout and improve overall well-being. Three factors have been isolated: first, emotional exhaustion; second, reduced personal fulfilment; and third, impersonality. Moreover, this research sheds light on the management

dimensions of professional burnout among nurses in retirement homes, emphasising the need for proactive strategies to address its far-reaching consequences. By quantifying the economic impact, healthcare organisations can better strategise resource allocation and implement cost-effective interventions to enhance workforce well-being, ultimately fostering a sustainable and economically resilient healthcare environment.

**Author Contributions:** Conceptualization, L.L. and S.G.; methodology, L.L. and S.G.; software, L.L.; validation, V.Š., Y.L.L. and S.G.; formal analysis, L.L.; investigation, R.F.; resources, Š.B.; data curation, L.L.; writing—original draft preparation, L.L. and S.G.; writing—review and editing, S.G., V.Š., R.F. and Š.B.; visualization, S.G.; supervision, Š.B.; project administration, V.Š.; funding acquisition, V.Š. and S.G. All authors have read and agreed to the published version of the manuscript.

**Funding:** This research received no external funding.

**Institutional Review Board Statement:** The study protocol was reviewed and approved by the Ethical Committee of Human Research at the Faculty of Health Sciences, University of Novo Mesto, Slovenia, working within the Committee for Research (approval No. FZV-145/2020, date 22 June 2020; track number 21/1).

**Informed Consent Statement:** Informed consent was obtained from all subjects involved in the study.

**Data Availability Statement:** Data is available on request.

**Acknowledgments:** Some tools for this research were used inside the research project CRP2023 V5-2331, financed by the Slovenian Research and Innovation Agency associated with the Juraj Dobrila University of Pula, Faculty of Economics and Tourism "Dr. Mijo Mirkovic" Pula, which co-financed the research.

**Conflicts of Interest:** The authors declare no conflicts of interest.

## Appendix A

**Table A1.** Pattern matrix MBI-HSS questionnaire (complete version).

| No. | Variable (Questionnaire Item) | Factor | | |
|---|---|---|---|---|
| | | 1 | 2 | 3 |
| 1 | At the end of work, I feel worn out. | 0.87 | −0.03 | −0.03 |
| 2 | I feel burnt out from work. | 0.83 | −0.06 | 0.10 |
| 3 | I feel tired when I get up to face a new workday. | 0.80 | 0.00 | −0.03 |
| 4 | I feel like I have no energy. | 0.78 | −0.04 | −0.01 |
| 5 | My work emotionally drains me. | 0.76 | −0.01 | 0.04 |
| 6 | My work frustrates me. | 0.70 | −0.09 | 0.00 |
| 7 | Working with people all day is tiring for me. | 0.69 | 0.08 | 0.06 |
| 8 | I get the feeling that I am doing too much. | 0.60 | 0.19 | −0.15 |
| 9 | Working with people results in too much stress for me. | 0.49 | 0.03 | 0.23 |
| 10 | I am effectively coping with my patient's problems. | 0.11 | 0.77 | 0.21 |
| 11 | I quickly established a relaxed atmosphere in patient relationships. | 0.07 | 0.66 | 0.00 |
| 12 | I feel full of energy. | −0.23 | 0.57 | 0.08 |
| 13 | It is easy for my patients to understand what and how they feel. | 0.10 | 0.53 | −0.05 |
| 14 | At work, I am very calm in dealing with emotional issues. | 0.12 | 0.50 | −0.16 |
| 15 | I have done many useful things at work. | 0.09 | 0.49 | −0.06 |
| 16 | I feel that I have a positive impact on people through my work. | −0.09 | 0.48 | −0.02 |
| 17 | I am happy when I work with patients. | −0.17 | 0.42 | −0.18 |
| 18 | I treat certain patients as impersonal objects. | −0.07 | −0.05 | 0.91 |
| 19 | I feel that patients blame me for their problems. | 0.07 | 0.10 | 0.62 |
| 20 | I have become colder to people since starting this job. | 0.05 | 0.02 | 0.53 |
| 21 | I do not really care what happens to my patients. | −0.10 | −0.06 | 0.50 |
| 22 | I am worried that my job is making me emotionally difficult. | 0.19 | −0.14 | 0.47 |

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
