# Peer review of "The Effect of Burnout Experienced by Nurses in Retirement Homes on Human Resources Economics"

_economies, doi:10.3390/economies12020033_

Round 1
Reviewer 1 Report
Comments and Suggestions for Authors
I have reviewed the paper entitled “The effect of burnout experienced by nurses in retirement homes on Human Resources Economics”. The paper has some shortcomings on the following issues. Therefore, I recommend a major revision.
1) The theoretical background of the study did not comprehensively deal with the issue of why there is a need for burnout impact experienced by nurses in retirement homes.
2) The literature review section is some kind of a hybrid section both providing some theoretical discussions and former findings. Therefore, the following revisions are required and must be deliberately made:
· Move all theoretical discussions to the introduction section.
· Summarize existing literature then critically evaluate the current state of the literature, and provide differences and contributions of the current study to the literature by unveiling the literature gap.
3) Try to visualize the data provided in Table 1 for ease of the read, thus, readers can make inferences at a glance.
4) Similar things stated in article 2 can be said for the discussion section as well.
· Move all theoretical discussions to the introduction section.
· Focus on only discussing obtained findings by comparing existing findings in the literature.
5) The current similarity level of the study is 15%. Authors should ensure that the revised article has no more than 20% similarity in the total and 1% from a single source
Comments on the Quality of English LanguageMinor revision is required.
Reviewer 2 Report
Comments and Suggestions for Authors
The first sentence of the abstract, it should be “The human resources economic implications of nursing burnout/ burnout amongst…”. The term, “economic implications of professional…” seemed to be not appropriate to describe the subject and topic of this study.
In the introduction, “the lack of work quality (QWL) is associated with higher levels of work-related, such as occupational stress…burnout…Burnout, as…” “Burnout should be emphasized instead of placing quality of work life. The first sentence may weaken the focus of burnout and burnout seemed to become one of the factors related to quality of work life.
The paragraph, “The paper is organized as follows..” can be deleted. The flow/ content/ structure does not need to be mentioned in the introduction. The aims and purposes of this study were not obviously mentioned.
In section 2. Materials and Methods, why did you mention the results there (i.e. 253 participants…40-49 years old). This should be the participants' characteristics.
Please separate the content in section 2 into the following sub-paragraphs: study design; outcome measures; and data analysis.
Please provide evidence to prove the sample size is reliable and feasible.
Also, limitations on small sample size were not mentioned.
Reviewer 3 Report
Comments and Suggestions for Authors
This article estimates the effect of burnout experienced by nurses in retirement homes on Human Resources Economics. Although it is very novel idea, the main purpose regarding nidification of the incidence and causes of burnout syndrome in nursing was not accomplished. The study has some key limitations.
- Small number of participants
- Participants and their inclusion of different occupational groups from multiple sites, albeit more representative, introduces greater heterogeneity of the sample and limits the generalizability of the results
- Lack of information about the baseline mental health status and previous history (depression/anxiety)
- No detailed participant characteristics, e.g. marital status, educational level, comorbidities, years in service in total, direct care of any COVID-19 patients, experience of COVID-19 symptoms, sufficient information from retirement health care authorities
In my point of view, the authors should include additional data before publication
Round 2
Reviewer 1 Report
Comments and Suggestions for Authors
The auhtors have adressed most of former queries. Although they stated that figure 1 is incorporated into the study, however, there was no figure in the paper. Therefore, there is a need for a minor revision to include regarding figure into the study.
Author Response
Thank you for your review and reply. The figure 1 is added now.
Reviewer 2 Report
Comments and Suggestions for Authors
In the responses, could you please clearly indicate what was added and deleted, and the reasons for doing these instead of stating "We have made the improvement.", "We have incorporated your comments and suggestions to improve the sections in question".
Reviewer 3 Report
Comments and Suggestions for Authors
After the revision of the arcticle, the main questions have been answered. The article can be published withoun further changes
Author Response
Thank you for your positive feedback.
Round 3
Reviewer 2 Report
Comments and Suggestions for Authors
Accepted
Author Response
Thank you for your positive reply.
